# An Observational Study in the Real Clinical Practice of the Treatment of Noninfectious Uveitis

**DOI:** 10.3390/jcm13051402

**Published:** 2024-02-28

**Authors:** Mar Esteban-Ortega, Martina Steiner, Cristina Andreu-Vázquez, Israel Thuissard-Vasallo, Alvaro Díaz-Rato, Santiago Muñoz-Fernández

**Affiliations:** 1Department of Ophthalmology, Infanta Sofía University Hospital, FIIB HUIS HHEN, Universidad Europea, 28702 Madrid, Spain; alvaro.diazr@salud.madrid.org; 2Department of Rheumatology, Infanta Sofía University Hospital, FIIB HUIS HHEN, Universidad Europea, 28702 Madrid, Spain; martina.steiner@salud.madrid.org (M.S.); santiago.munoz@salud.madrid.org (S.M.-F.); 3Department of Medicine, Faculty of Biomedical and Health Sciences, Universidad Europea, 28702 Madrid, Spain; cristina.andreu@universidadeuropea.es (C.A.-V.); israeljohn.thuissard@universidadeuropea.es (I.T.-V.)

**Keywords:** uveitis, immunosuppressive therapy, biologic therapy, retrospective study, macular edema

## Abstract

Background: The aim of this study was to describe the characteristics of patients with uveitis associated with an immunologic or idiopathic disease that requires immunosuppressive treatment and the response to such treatments in real clinical practice. Methods: An observational, descriptive, longitudinal, and retrospective study of a cohort of patients diagnosed with noninfectious uveitis was performed. To assess the response to treatment, we evaluated the change in visual acuity, vitritis, and the presence of macular edema. Results: We included 356 patients. Overall, 12% required treatment with systemic corticosteroids, and 66 patients (18.5%) required immunosuppressive/biological treatment, with methotrexate being the most used (55%). Immunosuppressive drugs were used in 59 cases (in 56 patients, as the first choice of treatment and for 3 patients as the second choice after treatment with biologics). Treatment with biologics was the first choice in 10 patients out of 66 (15%), and 34 (48%) required them at some time during the disease, with adalimumab being the most commonly used. Thirty-five patients (53%) needed to switch drugs due to a lack of response to the first one. There were no differences between different drugs in the resolution of vitritis and improvement in vision. Conclusions: The use of systemic corticosteroids and immunosuppressive/biologics was necessary for a high number of patients with noninfectious uveitis. In our series, tocilizumab was significantly more effective in the resolution of macular edema.

## 1. Introduction

Uveitis is an ocular disease that refers to a diverse group of intraocular inflammatory diseases of the uvea (the iris, ciliary body, and choroid), vitreous humor, retina, and optic nerve with a relatively wide list of causative agents (infectious, secondary to systemic disease, masquerade syndrome uveitis, or purely ophthalmologic or idiopathic causes). It is classified into four categories: anterior, intermediate, posterior, or panuveitis. The most frequent is anterior uveitis, and in general, the most serious are posterior uveitis and panuveitis. It most commonly presents during mid-adulthood with economic impact over time. Uveitis is a major cause of ocular morbidity and a frequent cause of blindness (10–25%), depending on the series. Between 20% and 70% of patients with uveitis experience significant vision loss, up to 50% of patients have reduced visual function, and 10% to 15% may go blind [1,2]. Vision loss in patients with this disease seems to be most directly related to the duration, severity, and location of inflammation, as well as complications. The most frequent complication is cystoid macular edema (CME) [1,3]. This risk seems to decrease with early treatment, thus improving the quality of life as well as reducing the socioeconomic impact of the disease [3,4].

The goals of therapy for noninfectious uveitis (NIU) are to reduce inflammation and achieve complete remission, thereby mitigating or avoiding ocular complications, permanent cumulative damage, and long-term vision loss. Rosenbaum et al. reviewed the indicated treatments for uveitis [3]. The therapeutic scale for the treatment of NIU begins with corticosteroids (CSs), which can be topical in anterior uveitis (AU) or, in less severe cases, with periocular corticosteroids (subtenonally or in the floor of the orbit) or systemic treatment (oral or intravenous in more severe uveitis), immunosuppressive (IS) drugs (T-cell inhibitors or antimetabolites), or biologic therapy (BT) [5]. Intravitreal CSs are used to avoid the adverse effects associated with systemic therapies while maintaining their anti-inflammatory effect limited to the eye for up to 3 years, mainly in cases of unilateral uveitis. However, their use has been associated with the appearance of increased intraocular pressure and the appearance or progression of cataracts [6,7].

Although systemic CSs at high doses decrease or eliminate ocular inflammation immediately, the use of IS drugs is needed to control the inflammation to avoid the side effects and potential complications of steroids and to keep the patient stable [8]. Although complete remission is not always possible, the main use of these drugs is to reduce exposure to CSs, and they are recommended as CS-sparing drugs when inflammation is not under control within 3 months [8,9].

The IS drugs used for this purpose include antimetabolites such as azathioprine, methotrexate (MTX), and mycophenolate mofetil (MFM); T-cell inhibitors such as cyclosporine (CyA); and tacrolimus and alkylating agents (cyclophosphamide and chlorambucil) [3,10].

Biologic therapy using tumor necrosis factor blocking (anti-TNFα) drugs is an alternative to the use of IS drugs. Adalimumab (ADA) is approved for the treatment of noninfectious nonanterior uveitis [11,12]. Infliximab (INF) and ADA have shown similar efficacy in controlling inflammation in chronic noninfectious uveitis, and the use of INF has been recommended for patients with Behcet’s disease and in severe acute uveitis [13,14]. Tocilizumab (TOZ), used in interleukin-6 inhibitor biologic therapy, showed improvement in visual acuity (VA) and a reduction in vitreous inflammation in patients with uveitis. It was well tolerated, with no adverse effects, and it demonstrated efficacy in severe and refractory uveitis, as well as in the resolution of CME secondary to uveitis [15,16].

Although the efficacy of all these therapies in the treatment of uveitis has been demonstrated, there are no clear guidelines or protocols due to the variability in their presentation and the approaching difficulty of the NIU.

The objectives of this study are to describe the characteristics of patients requiring IS treatment and/or BT in a multidisciplinary uveitis unit and the response to these treatments in different diseases.

## 2. Materials and Methods

An observational, descriptive, longitudinal, and retrospective study was carried out involving 356 patients diagnosed with idiopathic uveitis or uveitis associated with autoimmune disease including anterior, intermediate, or posterior uveitis, and/or panuveitis who visited the multidisciplinary uveitis unit of Infanta Sofia University Hospital. These patients were seen at some point during the 10-year period of the study (from January 2011 to February 2022). The hospital has a catchment area of 333,564 inhabitants.

In the multidisciplinary uveitis unit, a rheumatologist and an ophthalmologist consult together.

The research protocol for this study was approved by the research committee of the Hospital Universitario Infanta Sofía on 15 March 2022.

For each patient included in the study, we recorded sex and age. Data on the characteristics of uveitis, such as anatomical location (anterior, intermediate, posterior, or panuveitis), number of episodes, laterality (unilateral or bilateral), and etiology, were also recorded [17,18].

Systemic treatment with CSs, IS drugs, or BT was carried out, and markers of inflammatory activity (the presence or absence of vitritis as well as CME findings detected on optical coherence tomography (OCT) and VA) were recorded [18,19,20] before and after treatment to assess response. The best corrected visual acuity was measured according to LogMAR (the result of visual acuity in the logarithm of the minimum angle of resolution). The presence of vitritis was measured according to the SUN scale, and the presence of CME was assessed using Multimodal Swept Source Optical Coherence Tomography (Triton, Topcon^®^, Topcon Healthcare, Barcelona, Spain) [2,3]. Systemic treatment was used in severe cases with poor response to topical treatment or with a high number of recurrences. For the decision on immunosuppressive therapy, we followed the recommendations of the Spanish Society of Ocular Inflammation [21,22].

Absolute and relative frequencies were used to describe qualitative variables. Means and standard deviations (or median and interquartile range, [Q1, Q3]) were calculated for quantitative variables, according to their behavior (assessed by Shapiro–Wilk tests). The demographic and clinical characteristics of patients who required treatment with IS drugs and/or BT and those who did not were compared using chi-square or independent-sample Student’s *t*-tests (or Mann–Whitney U) as appropriate.

The responses to the different treatments with IS drugs and/or BT and their respective confidence intervals were calculated and compared using square tests. Similarly, the proportion of patients with resolved vitritis and CME and the change in VA after the different treatments were compared between treatments. A significance level of 5% was established, and all analyses were performed with STATA BE v.17 (StataCorp, College Station, TX, USA) statistical tools.

## 3. Results

### 3.1. Characteristics of Patients with Noninfectious Uveitis (NIU)

Table 1 summarizes the sociodemographic, clinical, and treatment characteristics of the 365 patients with NIU seen at the multidisciplinary uveitis unit at the Infanta Sofia University Hospital between January 2011 and February 2022. Of those, 180 were women (50.6%), and the mean age at diagnosis was 42.8 ± 15.8 years. A single episode of uveitis was presented in 157 (44%) of all patients, and 271 (76%) presented unilateral involvement. In terms of anatomical location, 265 patients (74.4%) presented with an AU, and in terms of etiology, the most frequent were idiopathic (43.3%) cases and those associated with human leucocyte antigen (HLA)-B27-positive axial spondyloarthritis (SA). Topical CSs were used in 313 (88%), 43 (12%) required systemic CSs, 26 required oral CSs, and 17 required intravenous CSs. In 66 patients (18.5%), treatment with IS drugs and/or BT was required to achieve uveitis remission. Of those, 34 were female (51.5%); the mean age was 39 ± 13 years at diagnosis and 42 ± 13 years at the time of treatment initiation. The time from uveitis diagnosis to the initiation of treatment with IS drugs and/or BT was 2 years [RIC: 2–4]. In 23 patients, treatment with IS drugs and/or BT was initiated in the first episode (34.9%), in 5 patients, in the second episode, and for 38 patients (57.6%), in the third episode or more. In total, 35 patients had unilateral involvement (53.0%).

### 3.2. Need for IS Treatment and/or BT 

IS treatment and/or BT was started in 23 patients in the first disease episode (35%), in 5 patients, in the second episode, and in 38 patients (58%) in the third episode or more. Overall, 35 patients had unilateral involvement (47.0%), and 31 patients had bilateral involvement (53.0%).

The characteristics of patients with NIU (IS treatment and/or BT or no IS/BT, the number of episodes, laterality, anatomic location, etiology, and the use of systemic corticosteroids) are described in Table 1.

Significant differences were found between patients who required treatment with IS drugs and/or BT and those who did not in age, laterality, anatomic location, and need for systemic CS use. Patients with IS treatment and/or BT were younger (38 (30–47) vs. 42 (32–55) years; *p* = 0.027), more frequently had bilateral uveitis (53% vs. 19%; *p* < 0.001), had the disease diagnosed in lower percentage anterior (48% vs. 80%; *p* < 0.001), and needed CS treatment more frequently (36% vs. 7%; *p* < 0.001) than patients who did not receive IS treatment or BT.

MTX was the most frequently used drug as the first line of IS treatment (30 patients, 53.6%), followed by CyA (11 patients, 19.6%), sulfasalazine (SSZ) (8 patients, 14%), and MFM (7 patients, 12.5%). In three of these patients, it was necessary to switch to a second IS due to a lack of response to the first one (Figure 1).

In 10 patients, BT was the first line of treatment without previous IS treatment. In 34 of the 66 patients (48%), BT was required at some point in the disease. A total of 20 patients (58.8%) received ADA as their first choice of treatment, followed in frequency by INF (*n* = 5 patients, 14.7%). Five patients needed to switch to another biologic due to a lack of response to the first biologic, with a good response to it. In total, 35 patients (53%) needed to switch to another IS or BT in the absence of a response to the first drug used (Figure 1).

### 3.3. Response to Treatment

Visual acuity after treatment increased in 94% of the patients, improving vision after treatment by 0.3 (0.1, 0.4) units, with the change being different between the different IS treatments.

These results are shown in Figure 2, which excludes those patients who started with a visual acuity of 1 and for whom the SI was used for other reasons such as frequent recurrences of uveitis.

Of the 26 patients with vitritis who received initial IS treatment (9 with CyA, 42.3%; 11 with MTX, 42.3%; and 6 with MFMX, 23.1%), vitritis persisted in only 1 patient. Two patients with vitritis received BT initially (one ADA and one TOZ). Vitritis persisted in only one of these patients, with intermediate uveitis initially treated with MFM, whose treatment was changed to ADA, and in this case, the inflammation was resolved. Eleven patients presented CME. Nine of them were initially treated with IS drugs (three with CyA, three with MTX, and three with MMF), and for two, treatment directly started with TOZ. Tocilizumab was chosen as the first therapeutic option in two cases with macular edema, due to its severity. Edema persisted in the three patients treated with CyA and resolved in one of the three patients and two of the three patients treated with MMF and MTX, respectively. One patient with Harada syndrome required up to four treatments for the resolution of CME (first treatment with CyA, second with MTX, third with ADA, and fourth with TOZ, which led to the resolution of CME). We also found a patient with bilateral UI who developed CME despite being treated with MFM, switching first to INF with no response and finally to ADA, which did resolve the inflammation (Figure 3).

Macular edema resolved in all patients who were treated with TOZ (seven out of seven), while in the rest of the treatments, the response was 0–50%. TOZ demonstrated a better response against CME than the rest of the treatments (100% vs. 29%, TOZ vs. rest; *p*-value = 0.005).

## 4. Discussion

This study was carried out to investigate the need and efficacy of treatment with IS drugs and/or BT in patients with idiopathic or autoimmune disease-associated uveitis. A systematic literature review revealed that these treatments may be effective in NIU patients [23,24].

We conducted a descriptive study of the patients with NIU followed up in the multidisciplinary uveitis unit of our hospital. The results did not differ from those reported in other published series [25,26,27,28]. There was no significant difference in sex, the mean age at diagnosis was 42.8 years, and the most frequent cases were anterior (74.4%) as well as unilateral (76.1%) uveitis. Notably, 157 patients (44.1%) had presented during this time with only one outbreak of ocular inflammation. Regarding etiology, the most frequent cases were idiopathic (43.3%), followed by uveitis associated with HLA B27+ AE (76 patients, 21.4%). The time from uveitis diagnosis to the initiation of treatment with immunosuppressants and/or biologics was 2 years [1, 4].

Of those patients with uveitis who required treatment with CSs via the general route, 19 were remitted without the need for subsequent immunosuppression as it was a single outbreak. Systemic treatment was necessary in severe cases, i.e., in those with complications such as vitritis, vasculitis, and macular edema, as well as in patients with a high number of recurrences. Patients with anterior uveitis and more than three recurrences per year despite topical treatment were also candidates for immunosuppressive drugs [29]. Of the total number of patients, 66 were treated with IS drugs and/or BT. The number of patients treated with IS drugs was similar to or somewhat higher than in other studies [28,30] and less than in others [26]. The etiology that most needed this type of treatment was idiopathic (21 patients, 31.4%). Ten patients started treatment directly with BT without previous IS treatment, while 56 (84.9%) had an IS as their first treatment. A total of 35 patients (53%) needed to switch to another IS or to BT due to a lack of response, with MMF being the one that required the switch on more occasions (six of seven patients, 85.7%). The proportion of males and females and the number of ocular inflammation flare-ups was not different between patients treated with IMS and/or BT and untreated patients, but treated patients were younger (perhaps because there was a higher number of posterior uveitis, intermediate uveitis, or panuveitis in this age group). These findings are similar to those reported in the study of Millán et al. [26]; the proportion of patients with bilateral uveitis was also higher in this group.

In our study, the most used drug was MTX (55.4%), followed by CyA (16.6%), in agreement with other studies, in which MTX was considered a drug with good response and low cost [26,29,31]. Compared to MFM, both have a high success rate [22,30], 63% and 64%, respectively, although in our case, MMF was used on seven occasions, and it required a change in treatment due to a lack of response in six of them (85.71%): three patients with intermediate uveitis, two with birdshot uveitis, and one with idiopathic RAAU.

In general, most patients with AAU (anterior acute uveitis) are treated with topical ocular CSs, but a small percentage of them have frequent recurrences (three or more outbreaks a year), requiring systemic treatment. MTX significantly reduces the number of flare-ups and activity in patients with idiopathic or systemic-associated AU [29]. We used MTX as the first IS drug in 30 patients, and of those, 16 had RAU diagnosed as either idiopathic or associated with the systemic disease: One patient was switched to SSZ due to poor tolerance, and in six, it was necessary to switch to BT due to a lack of response. Etanercept (ETA) has demonstrated clinical efficacy in the manifestation of AD, with a lack of data on the efficacy of AU and even the paradoxical appearance of AAU after treatment with this drug [32,33], although in our series, two patients with AAU associated with B27+ AS underwent ocular remission with ETA. SSZ was used as the first immunosuppressant in seven patients with RAASU associated with systemic conditions, requiring changes in treatment only in three patients due to lack of response [34].

In 34 of the 66 patients (48.5%), it was not possible to control the inflammatory activity with classic IS treatment, so it was necessary to introduce BT (mainly ADA) at some point in the disease, in general obtaining a very good response, as in previous studies [11,12,24].

Inflammatory activity, as well as the patient’s improvement or not, was measured according to variables (VA, vitritis, and CME), similar to the works of other authors [2]. The presence of visual worsening or any of these markers of inflammation in the presence of CS treatment indicated the need to add an IS drug. If despite this, inflammatory activity or vision loss continued, it was switched to another IS drug, or BT was added. Regarding VA, excluding patients who previously presented maximum VA (in these patients, the use of IS treatment was due to other criteria such as a greater number of flare-ups and recurrences), 94% exhibited improvement in vision. If we consider only those patients who initially presented vision loss due to uveitis, the proportion of them who experienced improved VA was similar among the different IS treatments. Regarding vitritis, present in 28 (42.4%) of the 66 patients who were treated with IS drugs and/or BT, it persisted in only 1 of them (a case of idiopathic intermediate uveitis that did not respond to MFM but did respond to ADA). In the rest of the patients, there was a positive response to treatment, with no differences between the different types of drugs. A patient with Harada syndrome needed up to four drugs to achieve CME remission and VA recovery, perhaps due to the delay in diagnosis and its chronicity [35].

CME can be a complication of uveitis in any of its locations and etiologies and is the main condition associated with VA loss [1,2]; therefore, it may benefit from early management including CS or IS treatment and/or BT [36]. Treatments with CyA, MTX, or MMF have shown an 83% reduction in the development of CME in patients with birdshot uveitis, and ADA and INF are effective in the treatment of CME in different uveitis entities. The efficacy of TOZ, an antibody against the IL-6 receptor, has been demonstrated in cases of refractory uveitis [15,24,37,38]. Subcutaneous sarilumab (SAR) may provide clinical benefit in the treatment of posterior NIU, especially in eyes with uveitic ME [39,40]. In our study, SAR was used in one patient with IU, and in one with refractory ME without response, TOZ was used, which led to its resolution. Macular edema resolved in all patients who were treated with TOZ (seven out of seven), while in the rest of the treatments, the response was 0–50%. TOZ demonstrated a better response against CME than the rest of the treatments (100% vs. 29%, TOZ vs. rest; *p*-value = 0.005). The 95% confidence interval (CI) for the proportion of patients responding to treatment with TOCZ was (64–100%) and the 95% CI for the response in patients treated with other drugs was (13–53%). These confidence intervals did not overlap, again indicating a statistically significant difference. Although our series involved a small sample, we believe that tocilizumab can be considered a useful drug in the treatment of macular edema in uveitis. New studies with a larger sample size are necessary to demonstrate the efficacy of TOZ in the treatment of ME in uveitis.

## 5. Conclusions

The use of immunosuppressive drugs, either conventional or biologic, alone or in combination, achieved, at least in our study, the objective of treating patients with uveitis without inflammatory activity. It should be noted that, in our series, TOZ proved to be significantly more effective in the resolution of macular edema. Regarding the advantages of our study, we can highlight that we used a wide range of real clinical cases with different drug treatments and different degrees of response observed among them. Future studies with a sample of patients undergoing treatment with a higher number of IS drugs/BT would be interesting. On the other hand, the limitations of our study include its retrospective nature, which may lead to inaccurate and nonstandardized data collected from evaluation at irregular intervals as well as the presence of small groups of drugs having to be grouped, which may create biases regarding the individual effectiveness of each drug. The main problem of our study is the lack of references that have studied the overall efficacy of different treatments in a large series of patients with idiopathic uveitis or uveitis associated with autoimmune disease. There are a limited number of randomized clinical trials demonstrating the efficacy of immunosuppression in the treatment of uveitis. Although immunomodulators have demonstrated efficacy in patients with NIU in preventing recurrences and controlling ocular inflammation, further research is needed to adequately define the role of each immunomodulator in this population.

## Figures and Tables

**Figure 1 jcm-13-01402-f001:**
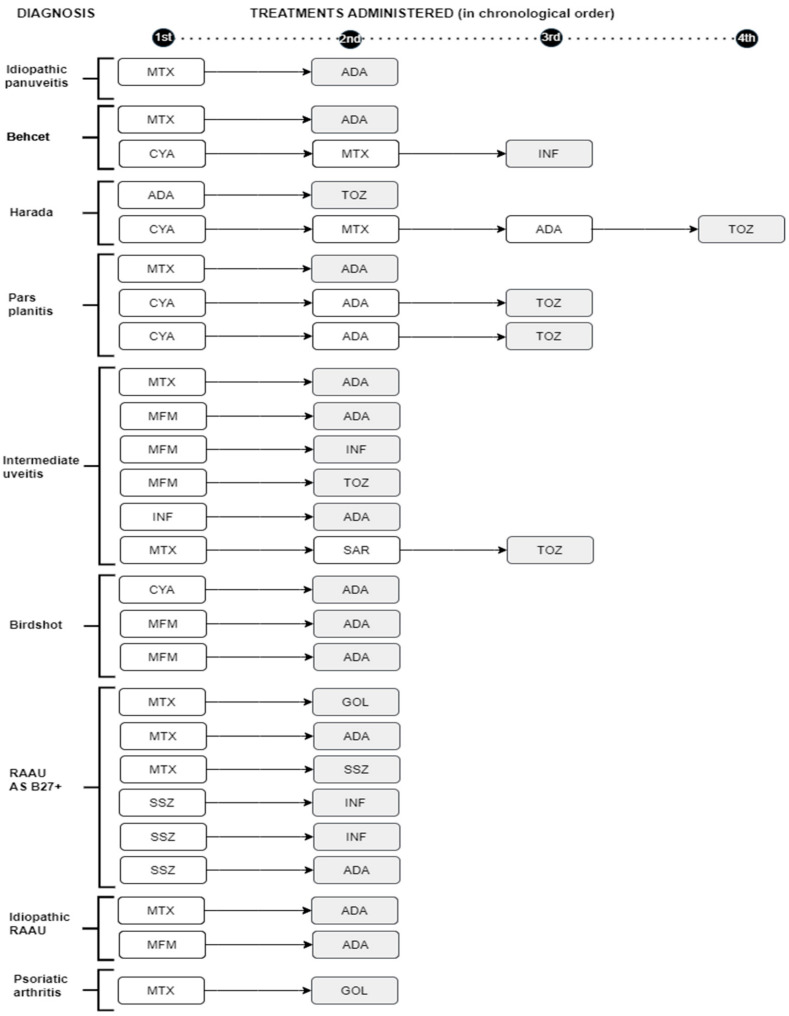
Timeline of immunosuppressive treatments and/or biologics administered in patients with noninfectious uveitis who required at least two treatments, according to diagnosis. Shaded: treatment that resolved the uveitis; RAAU: recurrent acute anterior uveitis; AS: axial spondyloarthropathy; MTX: methotrexate; ADA: adalimumab; MFM: mycophenolate mofetil; INF: infliximab; TOZ: tocilizumab; SAR: sarilumab; CYA: cyclosporine; SSZ: sulfasalazine.

**Figure 2 jcm-13-01402-f002:**
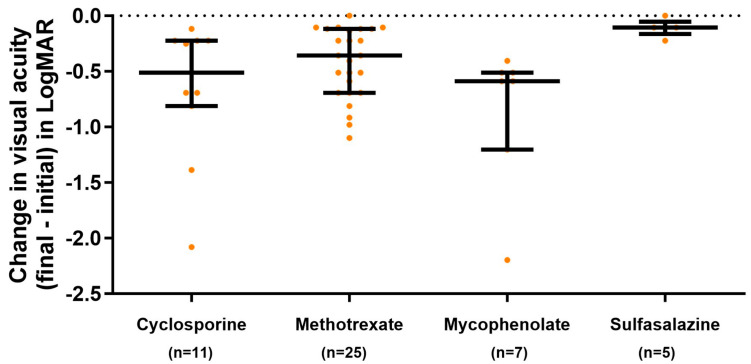
Change in visual acuity of the affected eye(s) in patients with noninfectious uveitis and visual acuity <1 before treatment according to the first immunosuppressant administered (median; Q1 and Q3). Patients who started with vision of 1 were excluded. Decimal scores were converted to logMAR using the formula logMAR = −log (decimal acuity).

**Figure 3 jcm-13-01402-f003:**
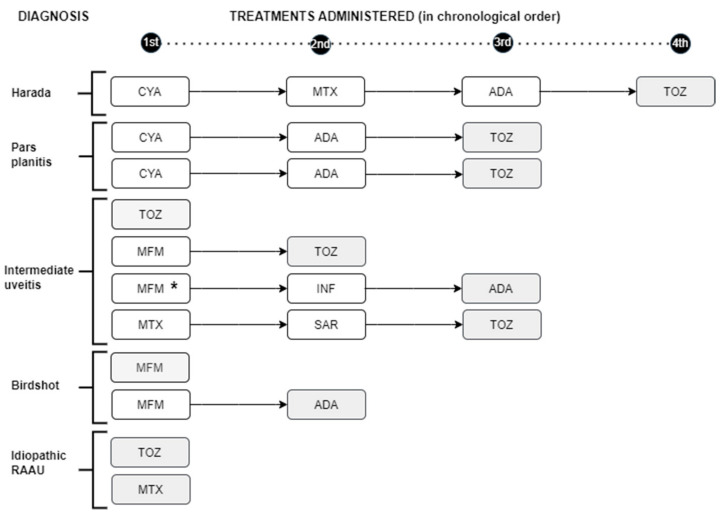
Timeline of immunosuppressive and/or biologic treatments administered in patients with noninfectious uveitis and macular edema, according to diagnosis. Shading: treatment that resolved macular edema. * Patient who developed macular edema while on MFM treatment. RAAU: recurrent acute anterior uveitis; CYA: cyclosporine; MTX: methotrexate; ADA: adalimumab; TOZ: tocilizumab; MFM: mycophenolate mofetil; INF: infliximab; SAR: sarilumab.

**Table 1 jcm-13-01402-t001:** Sociodemographic, clinical, and treatment * characteristics of all patients with UNI (*n* = 361) and comparison of untreated patients and those treated with IS/BT.

	Total	No TTO IS/BT	With TTO IS/BT	*p*-Value
	*n* = 356	*n* = 290 (81.5%)	*n* = 66 (18.5%)	
Female	180 (50.6)	146 (50.3)	34 (51.5)	0.864
Male	176 (49.4)	144 (49.7)	32 (48.5)	
Age (median [Q1Q3])				
At diagnosis	42.8 [31.5, 54.0]	42.0 [32.0, 55.0]	38.0 [30.0, 47.0]	0.027
At the start of treatment			42.5 [34.0, 50.0]	Not applicable
Time diagnosis–initiation TTO			2 [1, 4]	Not applicable
*n*. outbreaks				
1 outbreak	157 (44.1)	134 (46.2)	23 (34.9)	0.197
2 outbreak	35 (9.8)	30 (10.3)	5 (7.6)	
≥3 outbreak	164 (45.9)	126 (43.4)	38 (57.6)	
Laterality				
Unilateral	271 (76.1)	236 (81.4)	31 (47.0)	<0.001
Bilateral	85 (23.9)	54 (18.6)	35 (53.0)	
Anatomic Location				
Anterior	265 (74.4)	233 (80.3)	32 (48.5)	<0.001
Intermedia	34 (9.6)	22 (7.6)	12 (18.2)	
Posterior	28 (7.9)	18 (6.2)	10 (15.2)	
Panuveitis	29 (8.2)	17 (5.9)	12 (18.2)	
Etiology				
Idiopathic	154 (43.3)	133 (45.9)	21 (31.8)	Not applicable
AS HLA B27+	76 (21.4)	58 (20.0)	18 (27.3)	
Pars planitis	19 (5.3)	14 (4.8)	5 (7.6)	
Associated with HLA B27	17 (4.8)	16 (5.5)	1 (1.5)	
IBD	13 (3.7)	10 (3.5)	3 (4.6)	
AS HLA B27-	12 (3.4)	11 (3.8)	1 (1.5)	
Behcet	11 (3.1)	5 (1.7)	6 (9.1)	
Other white dot syndrome	8 (2.3)	7 (2.4)	1 (1.52)	
Posner	7 (2.0)	7 (2.4)	0	
Sarcoidosis	7 (2.0)	7 (2.4)	0	
Psoriatic arthritis	6 (1.7)	4 (1.4)	2 (3.0)	
Harada syndrome	5 (1.4)	3 (1.0)	2 (3.0)	
Birdshot uveitis	4 (1.1)	0	4 (6.1)	
APPC	4 (1.1)	4 (1.4)	0	
For drugs	4 (1.1)	4 (1.4)	0	
Intermedia-associated MS	3 (0.8)	1 (0.3)	2 (3.0)	
Associated Sjögren 1º	2 (0.6)	2 (0.7)	0	
Cogan’s syndrome	1 (0.3)	1 (0.3)	0	
Systemic lupus erythematosus	1 (0.3)	1 (0.3)	0	
TINU	1 (0.3)	1 (0.3)	0	
Undetermined	1 (0.3)	1 (0.3)	0	
Systemic CS treatment	43 (12.1)	19 (6.5)	24 (36.4)	
Orals	26 (7)	14 (4.8)	12 (18.2)	<0.001
Intravenous	17 (4.8)	5 (1.7)	12 (18.2)	

* Results are shown as *n* and (%) unless otherwise stated. TTO: treatment; AS: axial spondyloarthropathy; IBD: inflammatory bowel disease; APPC: acute posterior placoid chorioretinopathy; MS: multiple sclerosis; Sjogren 1º: Primary Sjogren TINU: tubulointerstitial nephritis with uveitis; CS: corticosteroid.

## Data Availability

The original contributions presented in the study are included in the article, further inquiries can be directed to the corresponding author.

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
