# Peer review of "An Observational Study in the Real Clinical Practice of the Treatment of Noninfectious Uveitis"

_jcm, 2024, doi:10.3390/jcm13051402_

Round 1

Reviewer 1 Report

Comments and Suggestions for Authors

Esteban-Ortega et al. reported the clinical data for non-infectious uveitis. The report was descriptive and partly involved treatment outcomes. Several points need to be clarified for the manuscript to be informative.

There was no description regarding inclusion and exclusion criteria for the study population. Did they involve all the uveitis patients with anterior, posterior, and pan- uveitis?

They seemed to describe that the study participants were those who consulted a single hospital during 2011 to 2022. Were they consecutive patients with non-infectious uveitis?

There were many patients who used systemic drug administration, which was one of their conclusions written in the abstract. It was not clear whether they included only severe cases who needed systemic treatment or not.

They reported that tocilizumab was effective to resolve macular edema, but the number of the patients with macular edema and treated by tocilizumab was only 7, and it would be hard to draw a conclusion from their observation. Also, the primary disease for the macular edema were not taken into account.

What were the application criteria for each treatment. Traditional drug might have been used after the patients had severe course and tocilizumab may have been used in the earlier phases because doctors may become used to apply biologics, recently.

Comments on the Quality of English Language

Mostly no problem.

Author Response

Thank you very much for taking the time to review this manuscript and for your comments. The corrections have been made and added to manuscript

COMMENTS AND SUGGESTIONS FOR AUTHORS

There was no description regarding inclusion and exclusion criteria for the study population. Did they involve all the uveitis patients with anterior, posterior, and pan- uveitis?

They seemed to describe that the study participants were those who consulted a single hospital during 2011 to 2022. Were they consecutive patients with non-infectious uveitis?

Yes, any type of uveitis was included: anterior, posterior, intermediate, panuveitis (line 93).

These patients were seen at some point during 10-year period (from January 2011 to February 2022) (line 95-96).

Observational, descriptive, longitudinal, and retrospective study of 356 patients diagnosed with idiopathic uveitis or uveitis associated with autoimmune disease both anterior, intermediate, posterior uveitis or/and panuveitis seen in the multidisciplinary uveitis unit of Infanta Sofia University hospital. These patients were seen at some point during 10-year period (from January 2011 to February 2022).

There were many patients who used systemic drug administration, which was one of their conclusions written in the abstract. It was not clear whether they included only severe cases who needed systemic treatment or not.

Systemic treatment was necessary in severe cases, in complications such as vitritis, vasculitis and macular edema and in patients with a high number of recurrences. Patients with anterior uveitis and more than three recurrences per year despite topical treatment were also candidates for oral immunosuppressants (line 238-242)

Of the uveitis that required treatment with CS via the general route, 19 were remitted without the need for subsequent immunosuppression as it was a single outbreak. Systemic treatment was necessary in severe cases, in complications such as vitritis, vasculitis and macular edema and in patients with a high number of recurrences. Patients with anterior uveitis and more than three recurrences per year despite topical treatment were also candidates for immunosuppressive drugs

They reported that tocilizumab was effective to resolve macular edema, but the number of the patients with macular edema and treated by tocilizumab was only 7, and it would be hard to draw a conclusion from their observation.

Although it is a small sample, the 7 patients treated with TOCILIZUMAB, respond while only 5 of the 17 treated with other drugs do. This difference in the proportion of responding cases is significant (p value = 0.05). The 95% confidence interval (CI) for the response proportion of patients treated with TOCILIZUMAB was (64%- 100%) and the 95% CI for the response in patients treated with other drugs was (13%-53%). These confidence intervals do not overlap, again indicating a statistically significant difference.

New studies with a larger sample size would be necessary to demonstrate the efficacy of Tocilizumab in the treatment of macular edema in uveitis. (Line 300-309, 316-317). The DISCUSSION of the manuscript adds the need to confirm these findings in future studies with a larger sample size. (Line 309-310)

Macular edema resolved in all patients who were treated with TOZ (7 of 7) while in the rest of the treatments, the response was 0-50%. TOZ demonstrated a better response against CME than the rest of the treatments (100% vs. 29%, TOZ vs. rest; p-value= 0.005). The 95% confidence interval (CI) for the response proportion of patients treated with TOCZ was (64%- 100%) and the 95% CI for the response in patients treated with other drugs was (13%-53%). These confidence intervals do not overlap, again indicating a statistically significant difference. Although it is a small sample in our series, we believe that tocilizumab can be considered a useful drug in the treatment of macular edema in uveitis. New studies with a larger sample size would be necessary to demonstrate the efficacy of TOZ in the treatment of ME in uveitis.

 Future studies with a sample of patients undergoing treatment with higher IS/BT would be interesting.

Also, the primary disease for the macular edema were not taken into account.

Figure 3 shows the diseases associated with non-infectious uveitis with macular edema.

What were the application criteria for each treatment? Traditional drug might have been used after the patients had severe course and tocilizumab may have been used in the earlier phases because doctors may become used to apply biologics, recently.

To decide the immunosuppressive therapy, we have followed the recommendations of the Spanish Society of Ocular Inflammation, published several years ago. To make it clearer in the text, we have introduced the following sentence at the end of the 4th paragraph in the material and methods section (line 114-116):

 For the decision on immunosuppressive therapy, we have followed the recommendations of the Spanish Society of Ocular Inflammation

Regarding tocilizumab, a total of 7 patients with macular edema were treated, in 5 of them after several failures with different immunosuppressants and only in 2 as the first therapeutic option due to the severity of the edema. These two patients responded satisfactorily. To clarify this point, we have added the following sentence at the end of the 4th paragraph of section 3.3 Response to treatment: "Tocilizumab was chosen as the first therapeutic option in two cases with macular edema, due to its severity."

Nine of them were initially treated with IS (3 with CyA, 3 with MTX and 3 with MMF) and 2 were started directly with TOZ. Tocilizumab was chosen as the first therapeutic option in two cases with macular edema, due to its severity.

I hope the errors have been corrected. Thanks again for your comments

Reviewer 2 Report

Comments and Suggestions for Authors

Overall, the paper is well written and the authors clearly state the limitations of their study which are sound and logical in any scientific measure. I have minor issues that will need to be addressed though:

In fig 2., the y axis is the final acuity - the initial acuity. From the figures presented, do the authors mean that the patients got worse acuities after treatment? If not, then the axis should be in the negative direction because on a LogMAR scale, a final acuity value if there is an improvement should have lower scores than the initial so a subtraction using the metric on the y axis should result in negative values.

Also, the abstract throws me away right from line 24 going. Can the authors be very clear in their presentation because as it stands, the interpretation gleaned from the abstract isn't as presented. First, they state that 66 patients received IS/biological treatment. If IS drugs were used in 59 cases, then it means you have about 7 left for the biological treatment. However, line 27 states that 10 started with biological treatment and these numbers do not add up. The authors should be more detailed and clear in the abstract for easy understanding.

Author Response

Thank you very much for taking the time to review this manuscript and for your comments. Please find the detailed responses below and the corresponding revisions/corrections highlighted/in track changes in the re-submitted files. 

Thank you very much for taking the time to review this manuscript and for your comments. Please find the detailed responses below and the corresponding revisions/corrections highlighted/in track changes in the re-submitted files.

COMMENTS AND SUGGESTIONS FOR AUTHORS

In fig 2., the y axis is the final acuity - the initial acuity. From the figures presented, do the authors mean that the patients got worse acuities after treatment? If not, then the axis should be in the negative direction because on a LogMAR scale, a final acuity value if there is an improvement should have lower scores than the initial so a subtraction using the metric on the y axis should result in negative values

Visual acuity has been measured with a decimal scale so that 1 is the maximum and 0.1 is the minimum. Therefore, going from 0.5 to 0.9, for example, is considered an improvement. I add in the manuscript that this was the way visual acuity was measured since I did not specify it. (Line 109-110)

Also, the abstract throws me away right from line 24 going. Can the authors be very clear in their presentation because as it stands, the interpretation gleaned from the abstract isn't as presented. First, they state that 66 patients received IS/biological treatment. If IS drugs were used in 59 cases, then it means you have about 7 left for the biological treatment. However, line 27 states that 10 started with biological treatment and these numbers do not add up. The authors should be more detailed and clear in the abstract for easy understanding.

Sorry, it was not clear enough. It has been changed in the abstract:

Immunosuppressive drugs were used in 59 cases (56 as first choice and 3 as second choice after biological treatment). Treatment with biologics was the first choice in 10 patients out of 66 (15%) and 34 (48%) required them at some time during the disease, with adalimumab being the most commonly used

I hope I have fixed the bugs. Thanks again for your comments